# COVID-19 Infection Induce miR-371a-3p Upregulation Resulting in Influence on Male Fertility

**DOI:** 10.3390/biomedicines10040858

**Published:** 2022-04-06

**Authors:** Heike Goebel, Barbara Koeditz, Manuel Huerta, Ersen Kameri, Tim Nestler, Thomas Kamphausen, Johannes Friemann, Matthias Hamdorf, Timo Ohrmann, Philipp Koehler, Oliver A. Cornely, Manuel Montesinos-Rongen, David Nicol, Hubert Schorle, Peter Boor, Alexander Quaas, Christian Pallasch, Axel Heidenreich, Melanie von Brandenstein

**Affiliations:** 1Institute of Pathology, Faculty of Medicine and University Hospital Cologne, University of Cologne, Kerpener Str. 62, 50937 Cologne, Germany; heike.goebel@uk-koeln.de (H.G.); alexander.quaas@uk-koeln.de (A.Q.); 2Clinic and Polyclinic for Urology, Faculty of Medicine and University Hospital Cologne, University of Cologne, Kerpener Str. 62, 50937 Cologne, Germany; barbara.koeditz@uk-koeln.de (B.K.); manuel.huerta@uk-koeln.de (M.H.); ersenkameri@gmail.com (E.K.); tim.nestler@uk-koeln.de (T.N.); timo.ohrmann@uk-koeln.de (T.O.); axel.heidenreich@uk-koeln.de (A.H.); 3Institute of Legal Medicine, Faculty of Medicine and University Hospital Cologne, University of Cologne, Melatengürtel 60/62, 50823 Cologne, Germany; thomas.kamphausen@uk-koeln.de; 4Klinikum Lüdenscheid, Institute for Pathology, University Hospital Cologne, University of Cologne, Paulmannshöher Straße 14, 58515 Lüdenscheid, Germany; johannes.friemann@uk-koeln.de; 5Department of Ophthalmology, Faculty of Medicine and University Hospital Cologne, University of Cologne, Kerpener Str. 62, 50937 Cologne, Germany; matthias.hamdorf@uk-koeln.de; 6Terasaki Institute for Biomedical Innovation (TIBI), 1018 Westwood Blvd, Los Angeles, CA 90024, USA; 7Excellence Center for Medical Mycology (ECMM), Department I of Internal Medicine, Faculty of Medicine and University Hospital Cologne, University of Cologne, Kerpener Str. 62, 50937 Cologne, Germany; philipp.koehler@uk-koeln.de (P.K.); oliver.cornely@uk-koeln.de (O.A.C.); christian.pallasch@uk-koeln.de (C.P.); 8Cologne Excellence Cluster on Cellular Stress Responses in Aging-Associated Diseases (CECAD), Chair Translational Research, Faculty of Medicine and University Hospital Cologne, Kerpener Str. 62, 50937 Cologne, Germany; 9Clinical Trials Centre Cologne (ZKS Köln), Faculty of Medicine and University Hospital Cologne, University of Cologne, Kerpener Str. 62, 50935 Cologne, Germany; 10German Center for Infection Research (DZIF), Partner Site Bonn-Cologne, Medical Faculty and University Hospital Cologne, University of Cologne, Kerpener Str. 62, 50937 Cologne, Germany; 11Department of Neuropathology, Faculty of Medicine and University Hospital Cologne, University of Cologne, Kerpener Str. 62, 50937 Cologne, Germany; manuel.montesinos-rongen@uk-koeln.de; 12The Royal Marsden NHS Foundation Trust, London SW3 6JJ, UK; david.nicol@rmh.nhs.uk; 13Department of Developmental Pathology, Institute of Pathology, University Bonn Clinics, University of Bonn, 53113 Bonn, Germany; schorle@uni-bonn.de; 14Department of Pathology, RWTH Aachen University, 52062 Aachen, Germany; pboor@ukaachen.de; 15Department of Urology, Medical University Vienna, 1090 Vienna, Austria

**Keywords:** COVID-19, SARS-CoV-2, androgen receptor, male infertility, miR-371a-3p

## Abstract

In December 2019, the first case of COVID-19 was reported and since then several groups have already published that the virus can be present in the testis. To study the influence of SARS-CoV-2 which cause a dysregulation of the androgen receptor (AR) level, thereby leading to fertility problems and inducing germ cell testicular changes in patients after the infection. Formalin-Fixed-Paraffin-Embedded (FFPE) testicular samples from patients who died with or as a result of COVID-19 (*n* = 32) with controls (*n* = 6), inflammatory changes (*n* = 9), seminoma with/without metastasis (*n* = 11) compared with healthy biopsy samples (*n* = 3) were analyzed and compared via qRT-PCR for the expression of miR-371a-3p. An immunohistochemical analysis (IHC) and ELISA were performed in order to highlight the miR-371a-3p targeting the AR. Serum samples of patients with mild or severe COVID-19 symptoms (*n* = 34) were analyzed for miR-371a-3p expression. In 70% of the analyzed postmortem testicular tissue samples, a significant upregulation of the miR-371a-3p was detected, and 75% of the samples showed a reduced spermatogenesis. In serum samples, the upregulation of the miR-371a-3p was also detectable. The upregulation of the miR-371a-3p is responsible for the downregulation of the AR in SARS-CoV-2-positive patients, resulting in decreased spermatogenesis. Since the dysregulation of the AR is associated with infertility, further studies have to confirm if the identified dysregulation is regressive after a declining infection.

## 1. Introduction

In December 2019, the first case of the new coronavirus, called SARS-CoV-2 was described in Wuhan, China [1,2] and subsequently spread all over the world; the disease caused by SARS-CoV-2 is called COVID-19. The most common symptoms of SARS-CoV-2 infection include fever, pneumonia, shortness of breath and cough. From its initial presentation as a respiratory tract infection, it can affect multiple other internal organs and the brain or eyes [3]. The involvement of multiple organs frequently results in hospitalization and intensive care unit (ICU) admittance, which is often associated with poor outcome in patients [3]. Furthermore, SARS-CoV-2 can infect healthy people as well as those with pre-existing medical conditions, e.g., diabetes and obesity [4]. The risk of infection and mortality also increases with higher age and male sex [4,5]. Since the first documentation of the virus, several mutations have developed with different levels of infectiousness [6].

Regarding the urogenital organs, it is discussed that the testes can act as a reservoir for viruses, and some of these viral infections are associated with the development of testicular cancer [7]. It is known that the entry point for SARS-CoV-2 is the angiotensin converting enzyme 2 (ACE2). In the testis, a high expression level of ACE2 is detected in Leydig and Sertoli cells [7,8].

Seminomas are germ cell testicular cancers (GCTC). The average age for developing GCTC is between 20 to 40 years [9]. The epigenetic predisposition for the development of testicular cancer can be traced back to the first trimester of intrauterine life [10]. Nevertheless, the development of testicular cancer depends on several different environmental factors [11,12]. Recently, Dieckmann et al. published that in the serum of GCTC patients, a significant increase in the microRNA (miR) 371a-3p can be detected [13]. This result led to the incorporation of this specific miR as a diagnostic marker and care for GCTC in German guidelines. However, an upregulation of this miR is also described in other cancer entities [14] and therefore it can be assumed that the miR acts as an additional risk factor for the development of cancer. Since the connection of testicular tissue with the miR-371a-3p is established and since miRNAs can be influenced by infections, we investigated whether SARS-CoV-2 can also influence the expression pattern of the miR-371a-3p [15].

MicroRNAs (miRs) are small non-coding RNAs which either bind to the 3′ UTR of target messenger RNAs (mRNA), leading to a decrease in translation, destabilization and faster degradation or transporting the target mRNAs into P-bodies for storing [16]. It is known that the miR-371a-3p binds to the 3′ UTR of the androgen receptor (AR), leading to decreased protein levels [17]. A reduced AR level is frequently associated with infertility and testicular cancer development [18].

We hypothesized that COVID-19 could be a possible trigger for the development of infertility and therefore investigated if an associated increase in miR-371a-3p levels exists, resulting in the downregulation of the AR. We analyzed testicular formalin fixed paraffinized embedded (FFPE) tissue sections from patients dying with or as a result of SARS-CoV-2 infection and compared them with seminoma tissues, tissue from patients dying in the ICU with or without extracorporeal membrane oxygenation (ECMO) and biopsied inflamed testicular tissue (Table 1 for detailed information). Testicular tissue samples from autopsy patients (for detailed information see Table 2) and testicular tissue samples from patients without inflammatory history or cancer (Table 2) were used as further controls. We tried to distinguish whether the miR-371a-3p up- and AR downregulation is specific to COVID-19 infection or if there is a general reaction pattern in inflammation and/or intensive care therapy. All of the correlating control patients were of the same age so that age-related dysregulations of spermatogenesis could be excluded.

Furthermore, we evaluated the levels of miR-371a-3p serum samples of COVID-19 patients as a possible future biomarker for late onset complications.

## 2. Materials and Methods

### 2.1. Testicular Histological Changes

We analyzed testicular tissue fixed in 4% buffered formalin (Sigma-Aldrich, Taufkirchen, Germany) obtained either by surgery or at autopsy and grouped them into the following five categories: patients who died with a positive SARS-CoV-2 qPCR test (*n* = 32); patients without SARS-CoV-2 infection but in need of critical care at the time of death with or without ECMO (*n* = 4); patients with non-SARS-CoV-2 dependent inflammation, such as orchitis, who underwent orchiectomy (*n* = 9); patients who died from different causes without need of critical care (*n* = 6), and from surgically resected testicular tissue for causes other than inflammation or cancer (*n* = 3). Patients with seminoma either with metastasis (*n* = 5) or without (*n* = 6) served as controls for miR-371a-3p levels.

### 2.2. FFPE Tissue Samples

Testicular formalin fixed paraffin embedded (FFPE) tissue sections from patients who died as a result of COVID-19 (*n* = 32) were compared with testicular autopsy tissue from patients who received intensive care with/without ECMO (*n* = 4) and surgically resected inflamed testicular tissue (*n* = 9). We chose testicular tissue samples from autopsy patients (average 68.5 years, *n* = 6) as the control group for COVID-19 infected patients. Surgically resected testicular tissue samples from patients without a history of inflammation or cancer (*n* = 3) were included in the seminoma control group, with an average age 37.5 years consisted of seminoma tissues, either with (*n* = 5) or without metastasis (*n* = 6). (For further explanation please see Table 1). In our COVID-19-collective, no orchitis was noted. The COVID-19 samples were fixed for up to 10 days in 4% buffered formalin prior to the embedding procedure according to a safety protocol used for these specific autopsies. Other tumor entities were excluded in this study. Patients’ agreement was obtained (BioMASOTA, University Hospital of Cologne, file reference 12–163, informed consent or coroner approved). All procedures performed in studies involving human participants were in accordance with the ethical standards of the institutional and/or national research committee and with the 1964 Helsinki Declaration and its later amendments or comparable ethical standards.

### 2.3. Serum Samples

Serum samples were collected from patients with mild or severe COVID-19 symptoms (female and male). All patients signed the BioMASOTA protocol. The average age of females with mild symptoms was 67.12 ± 17.34 (*n* = 10), and with severe symptoms 59.16 ± 5.83 (*n* = 6), males with mild symptoms 62.1 ± 6.5 years (*n* = 11) and with severe symptoms 71.4 ± 8.72 years (*n* = 7). From 250 µL serum, miRNA and mRNA was isolated according to Qiagen miRNA or mRNA protocol (Qiagen, Hilden, Germany).

### 2.4. Light Microscopy and Immunohistochemistry of FFPE Tissues

The morphology of the FFPE tissues was examined using 1 µm sections stained with routine H&E. Spermiogenesis was examined and classified according to Sigg et al. [19,20]. Further tissue sections (3 µm thick) were immunohistochemically stained for the diagnostically used androgen receptor (AR, monoclonal mouse anti-human androgen receptor, clone AR441, Code M3562, Agilent Dako, Santa Clara, CA, USA) using a BOND immunostainer at a dilution of 1: 400 and citrate epitope retrieval. Semi quantitative grading of AR immunoreactivity was minimal (+), medium + and strong ++.

Normal testis samples as well as intratesticular structures such as rete testis or epididymis served as (internal) positive controls (int con). Negative controls for the AR receptor IHC are not a part of a diagnostically stained specimen since the antibody was extensively validated before use in the diagnostics panel (golden standard of a pathological institute).

### 2.5. Double Labelling of AR Immunohistochemistry of FFPE Tissues and by miR-371a-3p In Situ PCR

AR staining was performed as previously described. After the final IHC step, slides were stored in tap water and immediately in situ PCR was performed as described in Nuovo et al. [21], with slight differences as follows: For the digestion of the tissue was performed with Proteinase K was used. The RT-PCR reaction was done with a one-step kit (QPO15, Biocat, Heidelberg, Germany). The PCR conditions were as recommended by the company. After the PCR step, slides were washed, counterstained and mounted with DAPI medium.

### 2.6. Immunofluorescence of ACE2 Protein

The 3 μm thick FFPE sections were deparaffinized in 1 × 10 min at room temperature (RT) in xylene (Sigma-Aldrich, Taufkirchen, Germany), followed by 1 × 5 min in 100% ethanol (Sigma-Aldrich, Taufkirchen, Germany) and 1 min in 70% ethanol and rinsed with distilled water at RT. The specimens were digested with Proteinase K (Qiagen, Hilden, Germany) for 30 min at RT. Blocking was performed for 30 min in 3% milk in PBS (Gibco, Darmstadt, Germany) and slides were incubated for 1 h at RT with specific primary antibodies (ACE2, ab15348, Abcam, Cambridge, UK) 1:500 in 3% milk in PBS. Following washing with PBS, the sections were incubated for 1 h with a secondary antibody (Goat-anti-rabbit IgG-FITC, Santa Cruz, sc2359, Heidelberg, Germany). Then, 1:1000 in PBS at RT (rinsed with PBS). Counterstaining with the DAPI mounting medium (nuclear staining, Thermo Scientific, Schwerte, Germany) was performed and slides were cover slipped. Negative controls were performed by omitting the primary antibody.

### 2.7. Protein Extraction from FFPE

As described in Ikeda et al., the protein extraction from FFPE tissue was performed [22]. Then, 10 μm sections were incubated for 15 sec in Xylol, mixed, and centrifuged for 2 min at full speed at RT. Furthermore, 100% ethanol was added for 2 min to the pellet and mixed, after which it was centrifuged for 2 min at full speed. The supernatant was discarded and the pellet was air dried. They were incubated for 20 min at 100 °C in 40 μL of RIPA buffer (Santa Cruz, Heidelberg, Germany), followed by an incubation period of 2 h at 60 °C. The samples were centrifuged at full speed at 4 °C for 20 min and stored at −80 °C until further use. As previously described, protein quantification was performed [23].

### 2.8. ELISA

To check for the AR of the autopsy tissue by a different technique, an ELISA was performed in non-coated 96 well plates (Brand GmbH & Co KG, Wertheim, Germany). A total of 1 µg of each patient sample was incubated in 50 µL PBS for 2 h at RT. The washing step was performed thrice with 1× PBS followed by an incubation step with an androgen-specific antibody (1:500) (Agilent Dako, Clone AR441, M3562, Madrid, Spain) in the test plate and ß-actin (1:500) (Santa Cruz, SC 47778, Heidelberg, Germany) in the control plate for 1 hr at RT. Afterwards, a 3× washing step was followed by incubation with the secondary antibody (Advansta, R-05071-500 goat anti mouse HRP conjugated, San Jose, CA, USA) (1:5000) for 1 hr at RT. After washing thrice, 50 µL substrate solution (Invitrogen, Schwerte, Germany) was added for 15 min and the reaction was stopped with a 50 µL stop solution (Bethyl Laboratories. Inc., Montgomery, TX, USA). The outread of the ELISA was performed at 450 nm.

### 2.9. miR Extraction from FFPE Tissue and RT-PCR

Furthermore, miRNA extraction from FFPE tissue was performed according to the miRNeasy FFPE kit (Qiagen, Hilden, Germany). For RNA quantification, NanoDrop technology was used.

The cDNA was obtained from 100 ng of RNA using MystiCq microRNA Primers (Sigma-Aldrich, St. Louis, MO, USA) and the miRScript miRNA reverse transcription kit (Qiagen, Hilden, Germany), according to the manufacturer’s protocol.

### 2.10. Quantitative Real-Time PCR (qRT-PCR)

The qRT-PCR was performed as previously described [24,25]. All samples were normalized to 5 s rRNA (miR-371a-3p) and to β-actin (AR wild type and AR TC as the reference gene). In order to calculate the relative fluorescence the ΔΔ-CT method was used, as outlined in User Bulletin 2 (PE Applied Biosystems, Darmstadt, Germany). Primer information is as follows: All PCRs were performed at 50 °C and 40 cycles. 5s rRNA: Forw 5′-GGCCAUACCACCCUGAACGC-3′, miR-371a-3p: Forw 5′-CACCGCGGTAA CACTCAAAAGAT-3′, ß-actin: Forw 5′-TTGGCAATGA GCGGTTCCGCTG-3′, Rev 5′-GACAGCACTGTGTTGCGTA-3′, AR: Forw 5′-CGGAAGCTGAAGAAACTTGG -3′, Rev 5′-ATGGCTTCCAGGACATTCAG -3′.

### 2.11. Statistical Analysis

For the statistical analysis, GraphPad Prism 5 (v5.0, San Diego, CA, USA) program was used and the analysis of variance (ANOVA) was performed. Significant differences were indicated by stars (* *p* < 0.05, ** *p* < 0.01, and *** *p* < 0.001).

## 3. Results

### 3.1. Immunofluorescence ACE2 in Testicular Tissue from Healthy Donors and COVID-19 Patients

To visualize the level of the ACE2 Receptor, the entry point of SARS-CoV-2-, FFPE tissue sections were subjected to immunofluorescence staining. In Figure 1 ACE2 expression is depicted in tissue samples of COVID-19 patients, autopsy samples and testicular biopsy samples (TESE). In all three samples, the expression of the ACE2 receptor was detected.

### 3.2. Histology and Expression of the Androgen Receptor in Testicular Tissue

In 75% of all specimens from COVID-19-positive patients, reduced spermatogenesis was detectable (Figure 2C) compared to COVID-19-negative patients (Figure 2A), inflammatory changes (Figure 2E) or previous intensive care controls (Figure 2G). For a detailed description see Table 2. In the two vaccinated patients who died with SARS-CoV-2, a normal spermatogenesis (yellow arrowheads) was detectable as well as the expression of the AR (Figure 2I,J). For evaluation of spermatogenesis, we used the diagnostically used classification by Sigg et al. [19].

Recently, a correlation between ACE2 and the AR was described [26]. Therefore, IHC-stainings of testicular tissues were performed (Figure 2B,D,F,H,J). The testicular tissues from COVID-19 negative patients (Figure 2B), inflammatory changes (Figure 2F) or previous intensive care controls (Figure 2H) were positive for the AR protein (indicated by the brown nuclei (black arrowheads, Figure 2B) albeit diminished in F and H. Samples from a patients with COVID-19 were negative for AR (Figure 2D). To address a possible degradation of the AR protein due to autolysis or prolonged fixation, we also examined the presence of AR by ELISA and demonstrated the protein in the autopsy cases also with a reduction in COVID-19 cases (Figure 2L). Since the same amount of protein was used for all samples (2 µg), a significant reduction (*p* < 0.001) could be detected in COVID-19 compared to the corresponding control (autopsy). A reduction of ±30% was notable in COVID-19 samples.

### 3.3. AR IHC Followed by miR-371a-3p In Situ PCR

To show the localization of the AR as well the as the miR-371a-3p and the co-regulation of both, IHC followed by in situ PCR was performed. Since the AR is a target of the miR-371a-3p, as published by Fletcher et al. [17], the miR levels should be low in samples where the AR is present and vice versa. As depicted in Figure 3, in samples (controls) showing high AR levels, the miR-371a-p expression is reduced. In COVID-19 samples where the AR levels were reduced, the miR-371a-3p was significantly upregulated. Figure 4 presents the results of the three independent experiments.

### 3.4. Expression of miR 371a-3p in Testicular Tissue Samples and Serum from COVID-19 Patients

The previously described finding of decreased AR levels in COVID-19 patients’ cases compared to patients within the same age group indicates that a possible Sars-CoV-2-induced mechanism is involved. We therefore examined the miR-371a-3p expression and found a significant overexpression in the COVID-19-positive tissue samples (Figure 3) but not in the control samples collected from patients who were treated in the ICU/with ECMO treatment or in tissue sample showing inflammation (Figure 5). None of the COVID-19-positive patients had been diagnosed with tumors (of any kind, as listed in Table 2).

## 4. Discussion

### 4.1. COVID-19 and Infiltration of Testes

Since the first documentation of COVID-19 cases in December 2019, different organs have been associated with SARS-CoV-2 infection apart from the respiratory system [1]. The association between COVID-19 and decreasing sperm quality was observed [27,28].

So far, we know (i) that testis, an immunological privileged organ, can serve as a reservoir for viruses [7]. For example, the Zika virus is detectable in ejaculates even after the infection is over [29]. (ii) In the testis, a high level of the ACE2 protein—the target protein of SARS-CoV-2—could be detected but with a decrease in older men [27,30] and it correlates with the AR in these testes. (iii) The spermatogenesis is reduced in older men [31]. As the analyzed COVID-19 patients were 73 years old on average, it is possible that the reduced spermatogenesis is due to the increased age. Nevertheless, in the control group from patients without COVID-19, a slightly higher spermatogenesis was seen by using the Sigg-classification.

Bhowmick et al. published a correlation between the AR and ACE2 [32]. The AR is a promotor for the transmembrane protease serine 2 (TMPRSS2), which is a co-receptor necessary for the internalization of SARS-CoV-2 [33]. Interestingly, at least in the lungs, a downregulation of ACE2 was observed following infection with SARS-CoV-2, which resulted in an increase in disease severity [34]. In the COVID-19 samples, the ACE2 was found to be present. However, as these patients suffered from an acute infection by SARS-CoV-2, it is possible that the levels decrease in testes as well after the infection.

It is postulated that the testosterone levels influence the degree of the SARS-CoV-2 infection and in some cases of severe infection, hypogonadism was detectable [27]. In general, females have lower levels of AR than older males (>70 years), this possibly explains the correlation that more males become infected than females [35]. Additionally, it is known that patients with AR deprivation therapy have a decreased risk of COVID-19 [36].

### 4.2. COVID-19 and Androgen Receptor (AR)

As mentioned, the testis can serve as reservoir for viruses [7] e.g., the Zika virus which is detectable in ejaculates after the symptomatic phase [29] and Epstein–Barr-Virus, which may play a role in tumor development [37]. Different groups published that SARS-CoV-2 infection is correlated to a decreased sperm quality or even a decreased fertility [38,39]. In older men, spermatogenesis is generally reduced [40,41]. This is most likely caused by a decreased protein level of AR. Since COVID-19-positive patients have a mean age of 73 years, the higher age could explain the reduced spermatogenesis. However, AR was detectable by IHC in age-matched controls so we speculate that a reduction in AR in COVID-19-positive patients could—at least in part—be due to infection with SARS-CoV-2. A reduction in spermatogenesis can be associated with a decreasing presence of the AR [41]. In most of our COVID-19 patients, a reduced spermatogenesis was detectable (Table 2) although the patients were older [37]. This would correlate well with our IHC, showing that in most COVID-19 testis the AR level is absent/reduced (Figure 2). However, it would be possible that due to the longer formalin fixation, the antibody binding was either compromised [42] or that the autolysis led to a degradation of the AR protein. This could be excluded, due to the ELISA against ß-actin and AR serving as an isolation/proof of principle control (Figure 2I,J). Figure 2J visualized the presence and a significant reduction in the AR level in COVID-19 patients, whereas ECMO-/autopsy specimens and inflamed testicular tissue indicated that IHC staining can be interpreted as specific and reliable. As mentioned, the AR mediates the expression of ACE2 and TMPRSS2, which explains that in most patients with AR deprivation therapy, a reduced risk of getting SARS-CoV-2 infections is notable [43]. However, the following question arises: why could we identify decreased AR levels in the analyzed COVID-19 patients. Is there another mechanism involved?

### 4.3. COVID-19 and miR-371a-3p Overexpression

It is shown that some virus infections are associated with a dysregulation of miRNAs [15]. The GCTC-specific miRNA, called miR-371a-3p, is responsible for the regulation of AR expression [44] and is also used as diagnostic marker for testicular cancer [13] according to the S3 guidelines of the testicular cancer diagnostics. Furthermore, it has also been published that the upregulation of miR-371a-3p in ejaculates can be used as a non-invasive marker for male infertility [45]. We therefore analyzed miR-371a-3p and were able to demonstrate a significant dysregulated expression of the miR-371a-3p (upregulation) in COVID-19 patients but not in the control groups (Figure 3). The dysregulation was even higher than in the seminoma control group (* *p* < 0.05). Since the miR-371a-3p is also upregulated in other cancer entities [14], we excluded tumor patients (apart from the seminoma group) in all patient groups (Table 2).

Until now, we have had no definitive timeline of SARS-CoV-2 presence or detectability in the body. Our findings of the increased miR level in postmortem COVID-19 testis and in the serum of living COVID-19 patients could be related either to the infection and might normalize over time or it can be the onset process that changes cell morphogenesis, resulting in reduced sperm production. Possibly, as a late outcome of COVID-19 infection and its upregulation of miR-371a-3p it could even be a risk factor for the development of a testicular germ cell cancer. Germ cell tumors arise from a primordial germ cell/gonocyte, which is arrested in development. These cells display a PGC-like expression pattern and remain within the testes environment. In young men during puberty, these precursors eventually start to proliferate to give rise to seminoma or embryonal carcinomas [46]. The exact cause of the developmental arrest as well as the nature of the events leading to the initiation of proliferation during puberty are only poorly understood [47]. It has been demonstrated that Germ Cell Tumors display elevated levels of miR-371a-3p. In fact, miR-371a-3p levels serve as a biomarker for detecting GCT [13]. In our study here, we have demonstrated, that an infection with SARS-CoV-2 leads to an increase in miR-371a-3p levels in serum. To exclude the described complications, a (long-time) follow up of patients after SARS-CoV2-infection would be a good strategy. Due to the low number of vaccinated patients in the study, we can only assume that the vaccine can prevent a reduced spermatogenesis and the correlating reduced expression of the AR receptor (*n* = 2). It has been shown in the literature that vaccination has no influence on sperm quality which correlates with our results [48].

### 4.4. AR Expression in COVID-19 Patients

The reduction in AR levels are possibly linked to the increased miR-371a-3p binding to the 3′ UTR of the AR, resulting in the downregulation of that protein [49]. Why should COVID-19 reduce AR which is necessary for the production of the docking and internalization receptors (TMPRSS2 and ACE2)? We speculate that the virus acts similarly to HI-Virus, which also downregulates these important receptors to prevent a super infection event [50]. Figure 4 depicts the results from three independent experiments, showing that the miR-371a-3p was significantly increased in COVID-19 samples, where the AR levels were reduced or absent and vice versa in the corresponding controls. Here, the TESE was used as high control, since in those samples no degradation of AR is expected as it could sometimes be in autopsy controls.

### 4.5. Detection of miR-371a-3p in Blood Samples

Since an upregulation of the miR-371a-3p was detectable in postmortem testicular tissue of the older male patients, we investigated the miR-371a-3p also in serum of younger COVID-19 male patients where it was also significantly increased (Figure 5). We postulate that in the setting of a SARS-CoV2-infection, this miR-371a-3p could be used to monitor a possible long COVID-like disease development, however, to be confident in this conclusion, further long-term investigations are needed. Since it also is a serum marker for the detection of testicular germ cell tumors [13], a urological follow-up examination should be included to prevent overseeing a seminoma by attributing the higher levels of miR-371a-3p only to COVID-19 and vice versa.

Nevertheless, a slight upregulation of the miR-371a-3p was also detectable in females since females also produce AR. The increase in testosterone results in higher levels of AR in females [51]. Higher levels of COVID-19 infections in older men compared to females could be explained by hormonal status.

## 5. Conclusions

We present here for the first time an association of SARS-CoV-2 infection involving the testes resulting in a downregulation of the AR by miR-371a-3p. This could influence the spermatogenesis and fertility after SARS-CoV-2 infection. Whether these increased levels of the miR-371a-3p are a long-term effect is still under investigation.

## Figures and Tables

**Figure 1 biomedicines-10-00858-f001:**
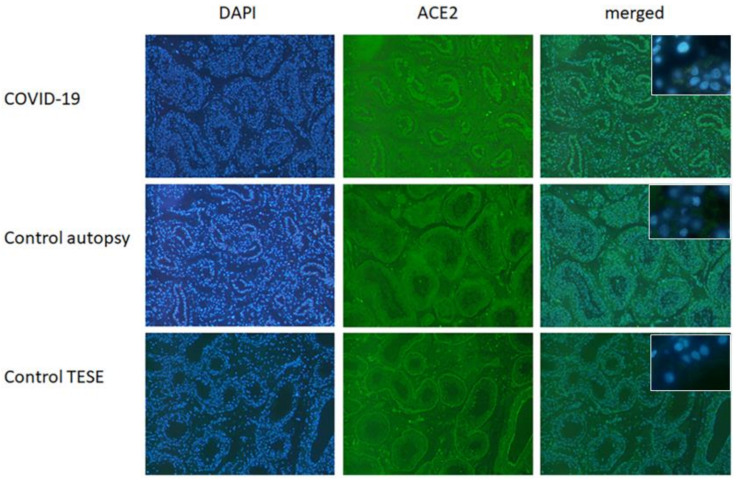
ACE2 in testicular FFPE samples from control (autopsy and testicular biopsy (TESE)) and COVID-19 patients. In all three depicted samples, ACE2 expression could be found. Shown here is a representative example out of five (40× magnification). Insert shows respective negative controls (63×).

**Figure 2 biomedicines-10-00858-f002:**
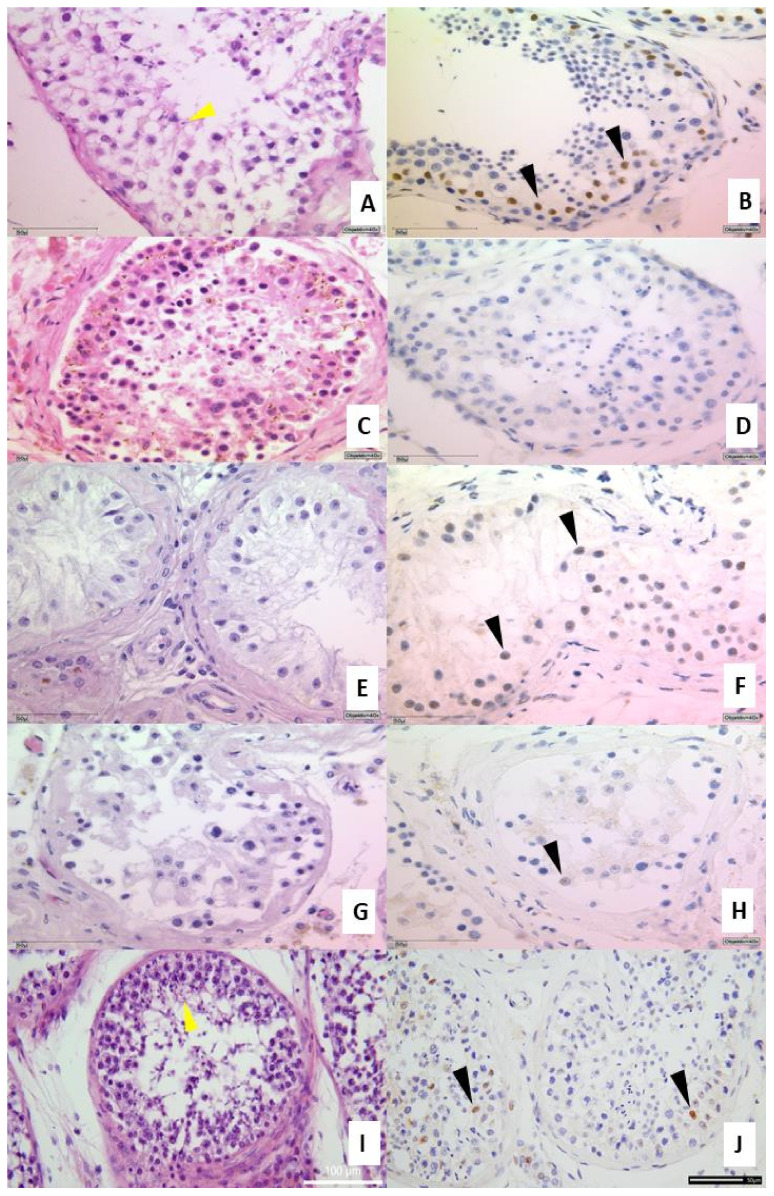
Upper part: IHC from testicular tissue samples. (**A**,**B**) present a slightly reduced spermatogenesis and normal AR expression in healthy control tissue sample (brownish marked nuclei, clone AR441) (**C**,**D**) Example of COVID-19 positive patient with no expression of the AR as well as a reduced spermatogenesis. (**E**,**F**) are examples of inflammatory changes control and (**G**,**H**) from autopsy of a case of previous intensive care treatment. (**I**,**J**) autopsy of vaccinated man dying of COVID-19. Figures exemplified a collective of each analyzed group used. Left side H&E (spermiogenesis yellow arrowheads), right side IHC for androgen receptor AR441 (black arrowheads), each at 400× magnification. Lower part: Androgen receptor ELISA from FFPE tissue. Total protein was isolated from FFPE tissue and 1 µg of total protein was used for the analysis by ELISA. Significant decreased AR values were found in testicular tissue samples compared with control group. *** *p* < 0.0001, ** *p* < 0.001, unmarked = ns. Control ELISA for ß-actin to control isolation quantity (**K**). Detection of AR protein in FFPE samples (**L**). Significant reduction in the AR levels in COVID-19 samples (±30%) compared to control autopsy and ICU non COVID-19 were detectable (*** *p* < 0.001).

**Figure 3 biomedicines-10-00858-f003:**
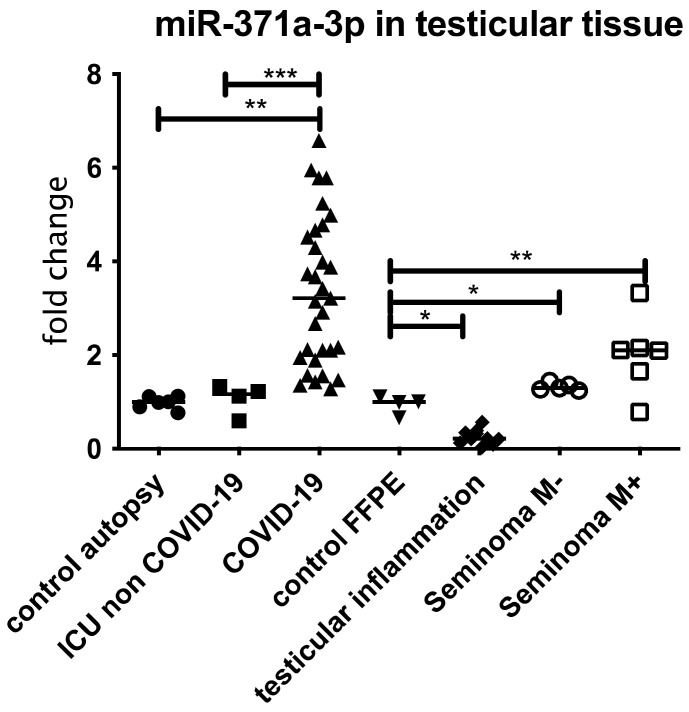
miR 371a-3p in FFPE testicular tissue samples from COVID-19 patients compared to patients with ICU/ECMO, inflamed testis and Seminoma patients without metastasis (M−) or with metastasis (M+). All samples were normalized to 5s rRNA and corresponding controls (for COVID-19 and ICU controls from autopsy patients and for inflamed testis and Seminoma controls from testicular surgery without inflammatory disease or tumor). An increased expression, comparable with the miR expression of Seminoma patients with metastasis was detectable. In case of ICU non COVID-19, Seminoma without metastasis and inflamed, a significant reduction compared to COVID-19 was detectable. *** *p* < 0.0001, ** *p* < 0.001, * *p* < 0.01, unmarked = ns.

**Figure 4 biomedicines-10-00858-f004:**
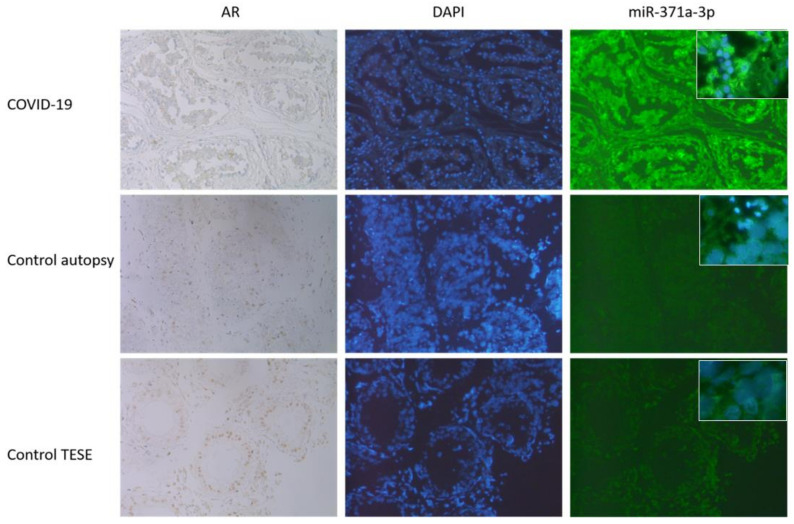
IHC followed by miR-371a-3p in situ PCR was performed. As it can be seen, those samples (controls, autopsy and TESE) showing high AR levels (here seen in brownish coloration of the nuclei, 40×) the miR-371a-3p expression is reduced (FITC staining right row, 40×, inserts 63×). DAPI staining was included to visualize the nuclei. In COVID-19 samples were the AR levels were reduced (nuclei appears blue, first row) the miR-371a-3p was significantly upregulated (last row). Here, results out of three independent experiments are presented.

**Figure 5 biomedicines-10-00858-f005:**
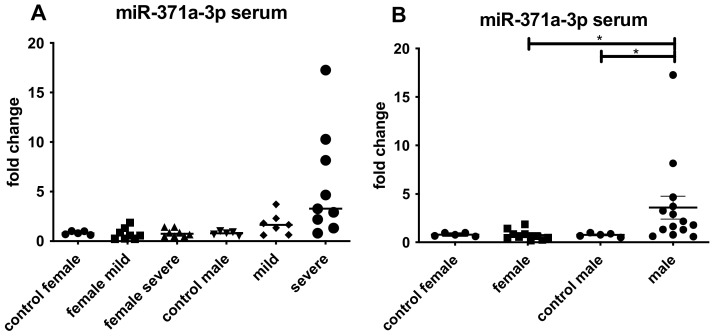
qRT-PCR results for the detection of miR-371a-3p in serum from patients with mild or severe COVID-19. (**A**) depicted the miR-371a-3p expression female and male. Upregulation was found in case of male patients either with mild or severe symptoms. (**B**) summarized all female and male cases showing the expression of the miR-371a-3p. Increase in male patients was detectable (* *p* < 0.01).

**Table 1 biomedicines-10-00858-t001:** Concluding the used samples and correlating controls.

Consecutively Autopsied COVID-19 Positive Male Patients: Spermiogenesis and miR-371a-3p	Control: Age Adjusted Autopsied Male Patients
	Control: testes biopsied (TESE) due to inflammation for exclusion of inflammatory process involvement
Living COVID-19 male patients: serum samples for miR-371a-3p	Control: Intensive Care/ECMO adjusted autopsied male patients for exclusion of therapy process involvement
	Control: seminoma for miR-371a-3p expression levels

**Table 2 biomedicines-10-00858-t002:** Patient characteristic. Abbreviations: n.d. = not determined, neg = negative, IC = internal control, GCNIS: germ cell neoplasia in-situ, nl= normal, (+) = slightly positive, + = positive.

	Age	BMI	AR IHC	Spermiogenesis	Sertoli Cells	Interstitial Area	Inflammation	Tumor/ GCNIS	Atrophy Class Acc. SIGG
	COVID-19								
	85	n.d.	neg	one testis: some spermatogonia and primary spermatocytes	autolytic changes	partial atrophy in one testes, the other complete atrophy	neg	neg	IIB
	70	n.d.	neg	normal spermiogenesis in both testes	autolytic changes	discrete interstitial edema	neg	neg	nl
	76	n.d.	neg	some spermatogonia and primary spermatocytes in both testes	autolytic changes	focal fibrosis, focal slightly thickened basement membrane of tubules	neg	neg	IIB
	80	37.1	neg	typical spermiogenesis in one testis, the other spermatogonia and primary spermatocytes	autolytic changes	interstitial edema	neg	neg	nl/IIB
	56	29	neg	single spermatogonia and primary spermatocytes	autolytic changes	slightly thickened basement membrane of tubules, discrete edema, focal interstitial fibrosis	neg	neg	IIB
	64	26.9	neg	single spermatogonia	autolytic changes	interstitial edema, focal interstitial fibrosis	neg	neg	IIC
	74	42.8	neg	single spermatogonia and primary spermatocytes	autolytic changes	medium interstitial fibrosis	neg	neg	IIB
	72	33.9	neg, IC +	reduced spermiogenesis	autolytic changes	discrete interstitial edema	neg	neg	I
	85	n.d.	neg	one testis: some spermatogonia and primary spermatocytes	autolytic changes	partial atrophy in one testes, the other complete atrophy	neg	neg	IIB
	70	n.d.	neg	normal spermiogenesis in both testes	autolytic changes	discrete interstitial edema	neg	neg	nl
	76	n.d.	neg	some spermatogonia and primary spermatocytes in both testes	autolytic changes	focal fibrosis, focal slightly thickened basement membrane of tubules	neg	neg	IIB
	80	37.1	neg	typical spermiogenesis in one testis, the other spermatogonia and primary spermatocytes	autolytic changes	interstitial edema	neg	neg	nl/IIB
	56	29	neg	single spermatogonia and primary spermatocytes	autolytic changes	slightly thickened basement membrane of tubules, discrete edema, focal interstitial fibrosis	neg	neg	IIB
	64	26.9	neg	single spermatogonia	autolytic changes	interstitial edema, focal interstitial fibrosis	neg	neg	IIC
	74	42.8	neg	single spermatogonia and primary spermatocytes	autolytic changes	medium interstitial fibrosis	neg	neg	IIB
	72	33.9	neg, IC+	reduced spermiogenesis	autolytic changes	discrete interstitial edema	neg	neg	I
	65	25.86	(+), IC +	normal spermiogenesis	autolytic changes	discrete interstitial edema	neg	neg	0
	72	31.14	neg	slight reduced spermiogenesis	autolytic changes	discrete interstitial edema	neg	neg	I
	47	20.76	pos, IC +	normal spermiogenesis	autolytic changes	discrete interstitial edema	neg	neg	0
	49	27	neg, IC (+)	sertoli only	autolytic changes	fibrosis	neg	neg	IV
	69	28.9	neg	reduced spermiogenesis	autolytic changes	discrete interstitial edema	neg	neg	IIa
	72	22.09	neg	sertoli only	autolytic changes	fibrosis	neg	neg	IV
	58	26.3	(+), IC neg	slight reduced spermiogenesis	autolytic changes	discrete interstitial edema	neg	neg	I
	76	31.6	neg, IC (+)	reduced spermiogenesis	autolytic changes	discrete interstitial edema	neg	neg	I-IIa
	83	27	(+), IC (+)	slight reduced spermiogenesis	autolytic changes	discrete interstitial edema	neg	neg	I
	58	24.7	neg	sertoli only	autolytic changes	interstitial edema, interstitial fibrosis	neg	neg	IV
	66	28.7	neg	sertoli only	autolytic changes	interstitial edema	neg	neg	IV
	55	23	neg, IC +	reduced spermiogenesis	autolytic changes	interstitial edema	neg	neg	III
	60	36.08	neg	sertoli only	autolytic changes	fibrosis	neg	neg	IV
	66	30.68	neg, IC (+)	reduced spermiogenesis	autolytic changes	edema	neg	neg	IIB
	73	23.67	neg, IC (+)	slight reduced spermiogenesis	autolytic changes	discrete interstitial edema	neg	neg	I
			(+), IC (+)	reduced spermiogenesis	autolytic changes	discrete interstitial edema, fibrosis	neg	neg	IIA/IIB
Mean	64	27							
	CONTROLS INFLAMMATION								
	57	n.d.	neg	extensive necrosis of tubular structures	neg	extensive necrosis of tubular structures	massive	neg	n.d.
	24	n.d.	+	spermatogonia and primary spermatocytes	pos	slightly thickened basement membrane, discrete edema	discrete	neg	IIB
	78	n.d.	focal (+)	partial tubules with Sertoli cells	pos	partial complete atrophy, focal slightly thickened basement membrane of tubules, focal interstitial fibrosis	massive	neg	IV
	85	n.d.	++	neg	autolytic changes	focal slightly thickened basement membrane of tubules, focal interstitial fibrosis	discrete	neg	IV–V
	75	n.d.	+, int con ++	neg	autolytic changes	focal slightly thickened basement membrane, focal interstitial fibrosis	focal	neg	V
	79	n.d.	++	some spermatogonia and primary spermatocytes and spermatids	pos	focal interstitial fibrosis and edema	discrete	neg	IIA
	79	n.d.	+	some spermatogonia and primary spermatocytes and spermatids	autolytic changes	focal slightly thickened basement membrane of tubules, focal interstitial fibrosis	discrete	neg	IIA
	81	n.d.	+	single spermatogonia and primary spermatocytes, no further spermiogenesis	autolytic changes	medium interstitial fibrosis	neg	neg	IIA
	44	n.d.	neg	neg	autolytic changes	focal interstitial fibrosis	massive	neg	n.d
Mean	66.8								
	AGE/AUTOPSY CONTROLS								
	61	n.d.	+	some spermatogonia and primary spermatocytes and spermatids	pos	focal slightly thickened basement membrane, focal interstitial fibrosis	neg	neg	IIA
	53	n.d.	+	reduced spermiogenesis		discrete interstitial edema	neg	neg	I
	63	n.d.	+	normal spermiogenesis		discrete interstitial edema	neg	neg	Nl
	76	23,2	neg	reduced spermiogenesis		discrete interstitial edema	neg	neg	I
	65	n.d.	neg	some spermatogonia and primary spermatocytes and spermatids	autolytic changes	focal slightly thickened basement membrane, focal interstitial fibrosi	neg	neg	IIA
	59		++	normal spermiogenesis		discrete interstitial edema	neg	neg	Nl
Mean	66.7								
	INTENSIVE CARE CONTROLS								
	67	39.1	neg	reduced spermiogenesis	pos	discrete interstitial edema	neg	neg	I
	55	n.d.	neg, int cont +	neg	autolytic changes	focal slightly thickened basement membrane of tubules, focal interstitial fibrosis	neg	neg	V
	57	44.1	neg	reduced spermiogenesis	pos	discrete interstitial edema	neg	neg	I
	76	norm	neg, int cont +	some spermatogonia and primary spermatocytes and spermatids	autolytic changes	focal slightly thickened basement membrane of tubules, focal interstitial fibrosis	neg	neg	IIA
Mean	63.75	41.6							
	SEMINOMA								
		n.d.	neg					pos	
	58	n.d.	neg					pos	
	36	n.d.	neg		ass			pos	
		n.d.	neg					pos	
	59	n.d.	n.d					pos	
		n.d.	neg					pos	
	51	n.d.	n.d.					pos	
	36	n.d.	neg					pos	
	27	n.d.	neg					pos	
	48	n.d.	neg					pos	
	51	n.d.	neg					pos	
	35	n.d.	n.d.					pos	
Mean	39.4								

## Data Availability

The data presented in this study are available on request from the corresponding author.

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
