# Peer review of "COVID-19 Infection Induce miR-371a-3p Upregulation Resulting in Influence on Male Fertility"

_biomedicines, 2022, doi:10.3390/biomedicines10040858_

Round 1

Reviewer 1 Report

Göbel et al., present a study where they evaluate different aspects of the infection by SARS-CoV2 in human testis. One of the more relevant aspects of the study is the inclusion of 28 COVID-19 patients, this is fantastic. The authors also included other patients (controls, tumors, etc.). Clearly, reproduction and COVID-19 is a hot topic and this kind of research it is very important, however some editions, experiments, etc, are required to make this manuscript more clear.

These are my suggestions

  1. The title of the paper is misleading, it does not represent any of the results included in the manuscript. This title is the perfect title for a review, but for this original publication  it looks like a "click bait"
  2. Introduction, it is necessary to add a sentence indicating that the virus is SARS-CoV-2 and the disease is COVID-19. Seems repetitive but  still a lot of confusion and given that the journal has a broad spectrum as a readers, it is necessary.
  3. Other important aspect it the lack of the references and the information already published by other of the effects in testis. There are at least 5 papers supporting this information and the should be included .
  4. it is not clear during the introduction why to study hmiR 371a-3p. authors talk about seminoma, then PGCs and then infection. It will be great if you explain first the role of this miR in infection first,  and do the link to your research
  5. Results, the quality of the pictures on figure 1 is very low in the PDF file, please include a high resolution pictures, also include your negative controls. How do you explain the lack of signal on the control patients? the fact that ACE2 expression can be affected by the SARS-CoV-2 infection does not mean that control should not have it. also this has been shown by other authors
  6. Figure 2, again the low quality of the pictures on the PDF made impossible to evaluate the results. Also did you perform quantification of the germ cell populations? I think that it will be very beneficial to have quantitative data. I think that the inclusion of a vaccinate patients it is amazing, but there is no real analysis of it? is there any differences compare to no vaccinated? it is more similar to control? what lessions do you see
  7. It is hard bur can you do a western blot to evaluate and confirm the protein levels on your samples
  8. Results from qpcrs in tissue and serum are  very clear, however what is the localization of this miRNA in the testis, is this coming from a specific cell type? It will be necessary to analyze this using a tandem approach of imumunodetection and miRNA detection such as RNAscope. 
  9. AR receptor is a target of miRN-371a, pleas show the drop of it on the testicular samples at least by IF
  10. minor commentS
    1. please check the F1 figure legend, it is not appropriate to say patients dying of corona! deceased? of samples for autopsies?
    2. just lie you did on figure 4, include all the data points in all the graphs 

Author Response

Dear reviewer,

We are thankful for your kind comments on our manuscript following our submission for publication.

We have re-looked at the manuscript and changed in line with your comments. Please see table attached.

The edited manuscript has a well-organized coding system that will guide you on the changes we have made.

Reviewer 2 Report

Manuscript ID: Biomedicines-1596245

Title: " COVID-19 - Implications for Future Fertility or Testicular 2 Germ Cell Changes?”.

 Authors : Göbel  et al .

Manuscript Type: Article

The manuscript of Gobel and colleagues reports a study aimed to investigate the correlation between spermatogenesis and fertility after SARS-CoV-2 infection. The authors  have studied  the

 the expression of miR-371a-3p and androgen receptor (AR) level in SARS-CoV2 positive patients

but however some specific criticisms should be addressed.

  1. In the paragraph” Testicular histological changes" (pag 3) I suggest adding The diagram showing the selection process of patient.

  1. In the paragraph” Histology and Expression of the Androgen Receptor in Testicular Tissue" (pag 5) and "Expression of miR 371a-3p in Testicular Tissue Samples and Serum from COVID-19 Patients" (pag 9)  I suggest adding in the text P value.

  1. In the section “Results” is not clear if the authors have studied the expression  of TMPRSS2 in Postmortem Testicular Tissue From Healthy Donors and Corona patients. This data would be important for the present study

  1. In table 1 it would be important to add information regarding the levels of the following hormones: testosterone (T) and luteinizing hormone (LH). In fact, COVID-19 patients showed a signifcant increase in serum LH level and a dramatic decrease in serum testosterone (T) to luteinizing hormone (LH) ratio, supporting the presence of a subclinical or compensated hypogonadism

  1. I think that in the future it will be necessary to replicate the study on a younger male population in order to confirm the reduced spermatogenesis in COVID-19 positive patients. This aspect should be better discussed in the discussion.

Author Response

(The authors gave the same response as above.)

Round 2

Reviewer 1 Report

The authors did a good work addressing  the questions and suggestions including the modification of the tittle.

Two main concerns,

a)In both figures 1 and 4, include the negative control (tissue just with secondary antibody). 

b) authors change the pictures but the resolution still low, I do not know if it is a problem of the pictures resolution or an artifact induced when the  files were exported to PDF.  In any case, it will be necessary to include an inset with a zoom in, showing a magnification of one tubule will help   the reader a lot. 

c) On figure 4, please do the zoom in and include the negative control. in the COVID patient seems that there is signal of the miR everywhere, normally miR signal are more like foci, how do you explain it? Also please include a zoom in

Author Response

Dear reviewer one,

thank you for your feedback. Please find attached our answers.

Comments and Suggestions for Authors

The authors did a good work addressing the questions and suggestions including the modification of the tittle.

Two main concerns,

  1. In both figures 1 and 4, include the negative control (tissue just with secondary antibody). 

Negative control is presented as insert in Fig 1.

Negative controls for the AR receptor IHC is not a part of a diagnostically stained specimen since the antibody was extensively validated before use in the diagnostics panel (golden standard of a pathological institute).

  1. b) Authors change the pictures but the resolution still low, I do not know if it is a problem of the pictures resolution or an artifact induced when the files were exported to PDF.  In any case, it will be necessary to include an inset with a zoom in, showing a magnification of one tubule will help the reader a lot. 

We added the picture again and proofed the resolution which was fine, even in the PDF file on our computer. If it does not show better resolution in your case, it may be another technical issue. We feel sorry for any inconvenience.

  1. c) On figure 4, please do the zoom in and include the negative control. In the COVID-19 patient seems that there is signal of the miR everywhere, normally miR signal are more like foci, how do you explain it? Also please include a zoom in

 We also inserted a zoom in in fig 4.

Since several different publications show that miR expression is significantly increased especially in tumor tissue, where the whole tumor area is stained, we assume that this is a general phenomenon. For sure we do not have tumor tissue here in our case but nevertheless, it can be excluded that this is a falsely positive signal, since negative controls and all other important controls were negative or at least slightly positive. According to our opinion the miR seems to be significantly overexpressed in nearly every cell in our examinate tissue samples (exemplified in picture 4, out 20 samples). This results correlate was well to the results of the qRT-PCR performed from those samples.  

  • https://www.researchgate.net/publication/223971533_Serial_selection_for_invasiveness_increases_expression_of_miR-143miR-145_in_glioblastoma_cell_lines/figures?lo=1
  • Detecting MicroRNA in Human Cancer Tissues with Fluorescence In Situ Hybridization
    ) https://journals.plos.org/plosone/article?id=10.1371/journal.pone.00535